# Antifungal Resistance in Clinical Isolates of *Candida glabrata* in Ibero-America

**DOI:** 10.3390/jof8010014

**Published:** 2021-12-26

**Authors:** Erick Martínez-Herrera, María Guadalupe Frías-De-León, Rigoberto Hernández-Castro, Eduardo García-Salazar, Roberto Arenas, Esther Ocharan-Hernández, Carmen Rodríguez-Cerdeira

**Affiliations:** 1Efficiency, Quality, and Costs in Health Services Research Group (EFISALUD), Galicia Sur Health Research Institute (IIS Galicia Sur), SERGAS-UVIGO, 36213 Vigo, Spain; erickmartinez_69@hotmail.com; 2Unidad de Investigación, Hospital Regional de Alta Especialidad de Ixtapaluca, Mexico City 56530, Estado de Mexico, Mexico; magpefrias@gmail.com (M.G.F.-D.-L.); eduardogs_01@hotmail.com (E.G.-S.); 3Sección de Estudios de Posgrado e Investigación, Escuela Superior de Medicina, Instituto Politécnico Nacional, Mexico City 11340, Mexico; estherocharan@hotmail.com; 4Departamento de Ecología de Agentes Patógenos, Hospital General Manuel Gea González, Mexico City 14080, Mexico; rigo37@gmail.com; 5Maestría en Ciencias de la Salud, Escuela Superior de Medicina, Instituto Politécnico Nacional, Mexico City 11340, Mexico; 6Sección de Micología, Hospital General Dr. Manuel Gea González, Mexico City 14080, Mexico; rarenas98@hotmail.com; 7Dermatology Department, Hospital Vithas Ntra. Sra. de Fátima, 36206 Vigo, Spain; 8Health Department, Campus Universitario, University of Vigo, 36310 Vigo, Spain

**Keywords:** *Candida glabrata*, antifungal resistance, azoles, echinocandines, ibero-america

## Abstract

In different regions worldwide, there exists an intra-and inter-regional variability in the rates of resistance to antifungal agents in *Candida glabrata*, highlighting the importance of understanding the epidemiology and antifungal susceptibility profiles of *C. glabrata* in each region. However, in some regions, such as Ibero-America, limited data are available in this context. Therefore, in the present study, a systematic review was conducted to determine the antifungal resistance in *C. glabrata* in Ibero-America over the last five years. A literature search for articles published between January 2015 and December 2020 was conducted without language restrictions, using the PubMed, Embase, Cochrane Library, and LILACS databases. The search terms that were used were “*Candida glabrata*” AND “antifungal resistance” AND “Country”, and 22 publications were retrieved from different countries. The use of azoles (fluconazole, itraconazole, voriconazole, posaconazole, isavuconazole, ketoconazole, and miconazole) varied between 4.0% and 100%, and that of echinocandins (micafungin, caspofungin, and anidulafungin) between 1.1% and 10.0%. The limited information on this subject in the region of Ibero-America emphasizes the need to identify the pathogens at the species level and perform antifungal susceptibility tests that may lead to the appropriate use of these drugs and the optimal doses in order to avoid the development of antifungal resistance or multi-resistance.

## 1. Introduction

In recent decades, the incidence of fungal infections has progressively increased, becoming a major public health problem worldwide [1]. Among these infections, candidiasis, which is caused by the yeasts belonging to the genus *Candida*, is the most common form of mycosis.

Within the genus *Candida*, there are more than 200 species, but only around 10% have been found to be associated with human infections [2], with *Candida albicans* being the most frequently isolated species; however, the isolation of non-albicans *Candida* species is more common because of the changes in the use of antifungal agents [2]. Among non-albicans species, *Candida glabrata* has been identified as one of the main opportunistic fungal pathogens in humans, ranking second to fourth as a causative agent of candidiasis, depending on the geographic region [3,4,5,6,7].

*C. glabrata* is a yeast that exhibits a greater phylogenetic relationship with *Saccharomyces cerevisiae* than with *C. albicans* and can colonize the genitourinary tract, intestine, and oral cavity of humans; however, under conditions of immunosuppression, it can cause mucocutaneous to invasive infections due to the presence of various virulence factors, such as the formation of thick biofilms [8,9]. Invasive infections often result in fatal outcomes [10]. The high rate of mortality reported in the cases of invasive *C. glabrata* infections has been related to both low intrinsic susceptibility and true resistance to fluconazole, which has been identified in a significant proportion of clinical isolates of this fungus [11]. Furthermore, certain isolates of *C. glabrata* can acquire cross-resistance to other azoles, including imidazoles and echinocandins, through exposure to these antifungal agents [9,12]. Resistance to echinocandins is increasingly frequent because this type of antifungal agents is used in patients with *C. glabrata* infections who did not respond previously to the treatment with azoles, which has resulted in the emergence of isolates resistant to echinocandins, as well as those resistant to both of these antifungal agents [9]. With respect to polyene antifungal agents, the isolates of *C. glabrata* are generally susceptible; however, there are some reports of reduced susceptibility to amphotericin B associated with a limited ergosterol content in the cell membrane [8].

The phenomenon of resistance to antifungal agents in *C. glabrata* is of particular relevance, as given the increase in the incidence of infections caused by this yeast, the therapeutic management of patients has become difficult because of the limited options of antifungal agents that can be used for these types of infections. The emergence of resistant *C. glabrata* isolates, particularly for azoles, has been reported in various regions worldwide, especially in the United States, Europe, Asia, and Australia, and the variability that exists at the inter- and intra-regional levels has been highlighted with regard to the susceptibility to antifungal agents [3,4,10,13,14]. For example, while in East Asia a low rate of resistance to azoles and high resistance to echinocandins have been reported, in Asia Pacific a low resistance to echinocandins has been reported [4,10]. These data on the variability in the resistance rates highlight the importance of understanding the epidemiology and susceptibility profiles of *C. glabrata* in each region. However, in some parts of the world, limited data are available in this context. For example, in Ibero-American countries, there is information regarding the increase in the frequency of infections, both superficial and invasive, caused by non-albicans *Candida* species, particularly *C. glabrata* [7,15,16]; however, information on antifungal resistance is scarce, because antifungal susceptibility testing has not been routinely performed.

In order to understand the resistance to antifungals in the genus *Candida*, and especially *C. glabrata*, we have worked for a long time on the analysis of the molecular mechanisms of resistance. This mainly focuses on those that are used more frequently, such as azoles, as well as the following areas:(a)Over-expression of membrane transporters: Here, the associated genes code for efflux pumps, and the first class involved in resistance to azoles is the superfamily of ATP-binding (adenosin triphosphate) cassettes (ABC) [17,18].(b)Altered ergosterol biosynthesis: Here, the associated genes are *ERG11/CYP51* where point mutations C108G, C423T, and A1581G have been observed. The ERG3 mutation involved is Q139A in Erg3p (C5 sterol desaturase enzyme), and ERG6 is the result of the formation of toxic sterols and not due to overexpression of the outflow pump [17,18].(c)Altered sterol import: Yeasts import sterols under anaerobic or microaerophilic conditions using the sterol importers Aus1p and Pdr11p [17,18].(d)Genome plasticity: In this mechanism, genomic variations must be considered, including loss of heterozygosity (LOH) and aneuploidy. In the case of *C. glabrata*, segmental rearrangements have been observed in the M and F chromosomes [17].

In polyenes, mainly amphotericin B, alterations made in vitro to the *ERG3* and *ERG6* genes decreased ergosterol levels and, likewise, resistance [17].

Point mutations of the *FKS1* gene have been observed in echinocandins. Among these are S629P, F625∆, and F625C. Likewise, the plasticity of the genome can increase resistance to these [17,18].

The inactivation of the enzymes cytokine permease, cytokine deaminase, and phosphoribosyl transferase cause an increase in the resistance to 5-Flurocytocin [17].

Therefore, in the present study, a systematic review was conducted to determine the antifungal resistance of *C. glabrata* in Ibero-America over the last five years.

## 2. Materials and Methods

A comprehensive literature search was conducted to evaluate the antifungal resistance of *C. glabrata* isolates. The search period spanned from 1 January 2015 to 31 December 2020. The search was conducted, without language restrictions, using the PubMed, Embase, Cochrane Library, and LILACS databases. The search words that were used were “Candida glabrata” AND “antifungal resistance” AND “Mexico” OR “Argentina” OR “Bolivia” OR “Chile” OR “Guatemala” OR “Spain” OR “Costa Rica” OR “Dominican Republic” OR “Brazil” OR “Cuba” OR “Ecuador” OR “El Salvador” OR “Nicaragua” OR “Panama” OR “Peru” OR “Paraguay” OR “Uruguay” OR “Venezuela” OR “Honduras” OR “Andorra” OR “Colombia” OR “Portugal.” The review was performed based on the preferred reporting elements for systematic reviews and meta-analyses (PRISMA) [19] (Figure 1).

## 3. Results

Of the 22 countries that constitute the Ibero-American region, 22 publications reported antifungal resistance in antifungal-resistant clinical isolates of *C. glabrata* during the last five years (Table 1). Most of these publications were from Spain, followed by Mexico, Brazil, Chile, Argentina, Cuba, Paraguay, Peru, and the Dominican Republic (Figure 1).

Resistance to azoles was reported in all the publications found, and only five publications reported resistance to echinocandins (Table 1). The resistance rate to azoles varied between 4.0% and 100%, while the resistance rate to echinocandins was found to be between 1.1% and 10.0%. With respect to azoles, *C. glabrata* was found to be resistant to fluconazole (MIC ≥ 4–128 mg/L) more frequently, followed by itraconazole (MIC 1 ≥ 4 mg/L), voriconazole (MIC ≥ 4 mg/L), posaconazole (MIC ≥ 2–8 mg/L), isavuconazole (MIC ≥ 0.5–4 mg/L), ketoconazole, and miconazole. Of the echinocandins, micafungin (MIC 0.03–0.5 mg/L) was the antifungal against which resistance was most frequently reported in *C. glabrata*, followed by caspofungin (MIC 0.084–0.25 mg/L) and anidulafungin (MIC 0.06–1 mg/L) (Figure 2).

These data correspond mainly to the isolates of *C. glabrata* obtained from different populations (women, children, hospitalized people, those with burns, etc.) with superficial (oral, vulvovaginal) or invasive (candidemia) candidiasis.

## 4. Discussion

Candidiasis is the most common form of mycosis reported worldwide [42]. Its main etiological agent, *C. albicans*, has been displaced by other species, for instance *C. glabrata*, which occupies the place between the second and fourth species, depending on the geographical region. This change in etiology is due, in large part, to the excessive use of antifungal agents in prophylactic treatments, which has also led to the development of resistance [43]. A distinctive feature of *C. glabrata* is its low intrinsic susceptibility to fluconazole; however, the emergence of isolates with a true resistance to fluconazole has been constantly registered in recent years, leading to a greater use of echinocandins and other azoles, resulting in the emergence of isolates resistant to some of these types of antifungal agents; and thus, multidrug-resistant isolates have emerged [44,45]. Antifungal resistance in *C. glabrata* has been shown to vary with the geographic region [46], which highlights the importance of studying the antifungal susceptibility patterns at the local and regional levels.

In this study, we analyzed the current situation (2015–2020) of antifungal resistance of *C. glabrata* in 22 countries that constitute the Ibero-American region. We found that only 41% of the countries (Argentina, Brazil, Chile, Cuba, Spain, Mexico, Paraguay, Peru, and the Dominican Republic) have published on the subject, with Spain and Mexico being the countries with most of the reports. This finding immediately indicates a lack of information on this study topic with respect to other geographical regions, such as the United States, Asia, and Europe, where most of the surveillance studies on antifungal resistance have been conducted, both on yeasts as well as on filamentous fungi [4].

In Ibero-America, *C. glabrata* is more resistant to fluconazole and other azoles (itraconazole, voriconazole, ketoconazole, miconazole, posaconazole, and isavuconazole) than to echinocandins (micafungin, caspofungin, and anidulafungin). The high resistance rate (4–100%) to azoles, particularly fluconazole, and the low resistance rate (1–10%) to echinocandins is consistent with the results in other regions [47]. For example, in various countries in the Asia-Pacific region, *C. glabrata* has been reported to have a 5.2% resistance rate to fluconazole and a low rate (1.7%) of resistance to echinocandins [4]. Likewise, in some European countries, such as Poland, resistance to fluconazole has been reported in up to 22% of isolates, but resistance to echinocandins is practically nil [48]. Conversely, countries such as Australia and Germany have reported high rates of resistance to fluconazole (22.8–38%) and echinocandins (17.1–48%) [3,13,14]. In African countries, the rates of resistance to fluconazole are variable (3–19%), but resistance to echinocandins is low [49,50,51], while resistance to fluconazole and echinocandins is rare in some regions of the United States [14]. However, it should be noted that the resistance rate that we find in Ibero-America depends on the cut-off points that have been used in each country. For example, in two studies conducted in Mexico and Spain, 32 mg/L was used as the cut-off point for fluconazole [32,39], while most studies used >64 mg/L. This discrepancy in the cut-off points may lead to a wrong interpretation in the definition of resistance, since 32 mg/L is considered a dose-dependent susceptibility, according to the document CLSI M27-A3. Likewise, the method used to determine the susceptibility to antifungals influences the definition of resistance, while Sensititre Yeast considers a resistance to fluconazole as MIC values ≥4 mg/L, and the broth microdilution method defines a sensitivity to fluconazole as values ≤8 mg/L.

Of the echinocandins, we found that micafungin was the agent exhibiting the highest resistance (Table 1), contrary to what was reported in Greece, where *C. glabrata* isolates tended to exhibit more resistance to caspofungin than to micafungin or anidulafungin [52].

With respect to other azoles, itraconazole ranked second in Ibero-America as the azole to which the clinical isolates of *C. glabrata* are most frequently resistant, in contrast to what has been reported in studies with isolates of *C. glabrata* from other countries, where the rates of resistance to fluconazole and itraconazole have been reported to be similar [53].

Without a doubt, we know that the appearance of resistance to antifungal agents in *C. glabrata* complicates the therapeutic management of infections caused by this fungus. Therefore, it is also important to be considerate about the percentage of isolates that present a dose-dependent susceptibility (DDS) in order to detect significant changes and guide antifungal therapy in a more effective way. During the study period in Ibero-America, it was found that a considerable proportion of *C. glabrata* isolates presented DDS, especially fluconazole, although DDS to itraconazole, voriconazole, flucytosine, and amphotericin B have also been reported [19,20,54,55]. Even in Venezuela, where no reports of resistance were found, in up to 50% of the clinical isolates of *C. glabrata*, DDS to fluconazole have been reported [56]. The high rate of *C. glabrata* isolates with DDS to fluconazole is consistent with that reported in different regions worldwide, such as Canada, Kuwait, Spain, Poland, Greece, and Jerusalem [57,58,59,60].

In this study, we did not find reports of resistance to polyenes in the clinical isolates of *C. glabrata*, which is consistent with what was reported in surveillance studies of antifungal susceptibility [61].

It is important to mention that the present study has at least two limitations: (1) reports were not found in all countries that constitute the Ibero-American region; and (2) the low number of isolates that were analyzed in each of the reports found. Clearly, this does not indicate that there is no antifungal resistance in most countries of the Ibero-American region. Given that the antifungal resistance exhibits an intra-and inter-regional variability, it is difficult to establish a trend regarding the resistance to fluconazole or echinocandins in the Ibero-American region. A problem associated with the lack of information may be that, in some places in this region, pathogens’ identification has not been conducted at the species level and far fewer antifungal susceptibility tests are routinely performed, which poses a major problem for decision making in refractory infections. For this reason, it is essential to conduct surveillance studies of antifungal resistance at the regional level, since the variability observed in different parts of the world prevents one from generalizing trends.

## 5. Conclusions

An increased resistance to azoles and echinocandins in *C. glabrata* is a serious problem in clinical settings worldwide. The scarce information on this subject in the Ibero-American region emphasizes the need to perform the identification of pathogens at the species level and to conduct antifungal susceptibility tests that lead to their appropriate use and the optimal doses in order to avoid the development of antifungal resistance or multi-resistance.

## Figures and Tables

**Figure 1 jof-08-00014-f001:**
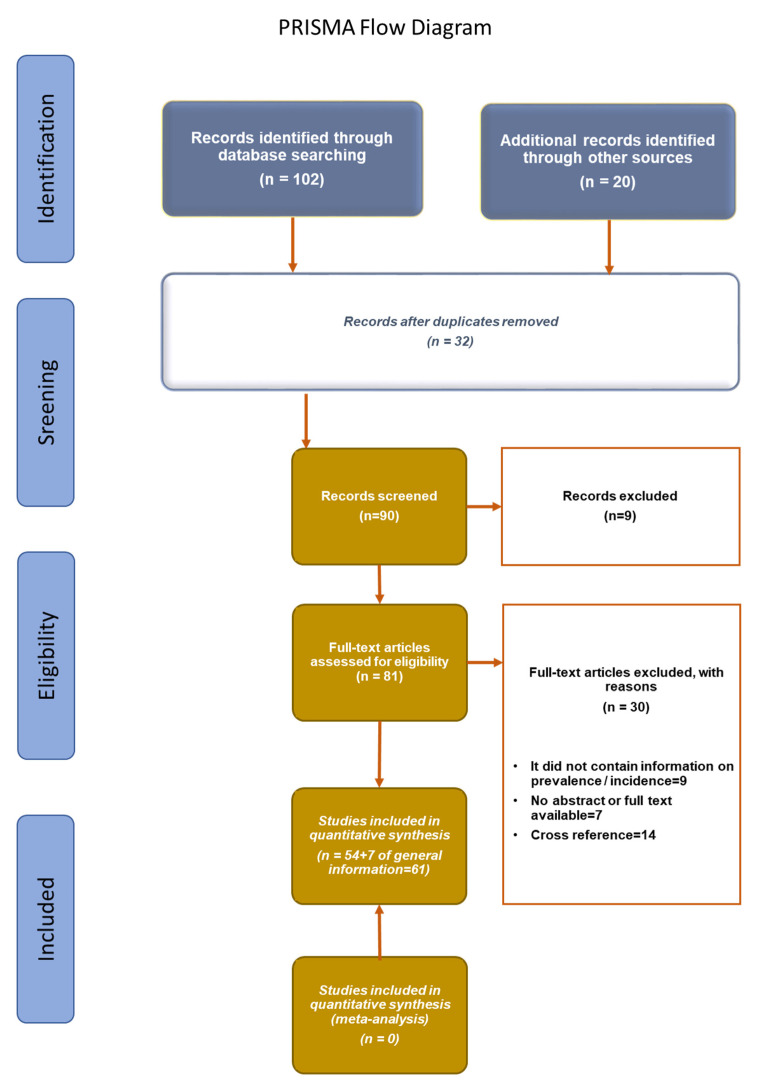
Flowchart of the different phases of the systematic review.

**Figure 2 jof-08-00014-f002:**
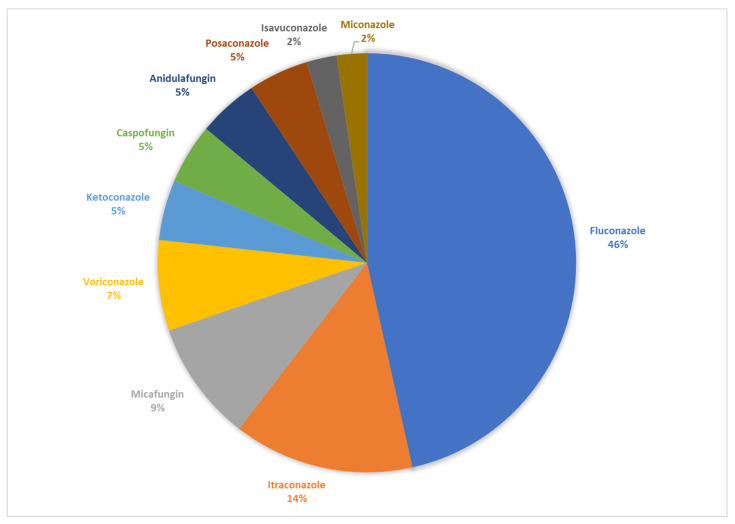
Antifungal resistance to *C. glabrata* in Ibero-American countries.

**Table 1 jof-08-00014-t001:** Antifungal resistance in clinical isolates of *C. glabrata sensu stricto* from Ibero-American countries.

Country	Type of Infection/Population	No. Isolates	Resistance Percentage (%)	MIC Antifungals	Years Studied	ASFT Method	Resistance Breakpoint	Reference
Argentina	Candidemia/Hospitalized 0–98 years	52	12.8	Fluconazole ≥ 64 mg/L	January 1998 to December 2013	Diffusion (CLSI M27-A3)	>64 mg/L	[20]
Brazil	Colonization/oral HIV/AIDS	14	14.3	Itraconazole ≥ 1 mg/L	January to May 2015	Microdilution Assay (CLSI M27-S4)	>1 mg/L	[21]
Brazil	Hospitalized different units	12	25.0	Voriconazole [NR] Fluconazole ≥ 64 mg/L Ketoconazole [NR]	September 2013 to May 2014	Diffusion 44-A (CLSI, 2004)	NR >64 mg/L NR	[22]
Chile	Candidemia	37	6.6 20.0 10.0	Fluconazole 8 mg/L * Itraconazole 0.5 mg/L Micafungin 0.25–0.5 mg/L	January 2013 to October 2017	Microdilution Assay (CLSI, M27-S4)	≥4 mg/L NR NR	[23]
Chile	Candidemia	3	100	Fluconazole ≥ 64 mg/L	Mach 2009 to August 2011	Diffusion (M44-A del CLSI (2004))	>64 mg/L	[24]
Cuba	Vaginal isolates	5	60.0	Itraconazole ≥ 1 mg/L	2015	Microdilution Assay (CLSI)	>1 mg/L	[25]
Spain	Invasive candidiasis	2	50.0	Fluconazole ≥4 mg/L *	January 2012 to December 2013	Microdilution Assay (CLSI)	≥4 mg/L	[26]
Spain	Candidemia	97	100	Fluconazole ≥4 mg/L *	April 2010 to May 2011	Microdilution Assay (CLSI M27-A3)	≥4 mg/L	[27]
Spain	Candidemia	14	50.0	Itraconazole 1 mg/L	January 2001 to December 2012	Microdilution Assay (CLS M27-A3)	>1 mg/L	[28]
Spain	Candidemia	94	10.6 1.1 1.1 1.1	Fluconazole 64 mg/L Micafungin 0.03 mg/L Caspofungin 0.25 mg/L Anidulafungin 0.06 mg/L	May 2010 to April 2011	Microdilution Assay—EUCAST (E. Def 7.1 and E. Def 7.2) y CLSI M27-A3	>64 mg/L >0.03 mg/L >0.25 mg/L >0.06 mg/L	[29]
Spain	Candidemia in burn patients	3	33.3	Fluconazole ≥4 mg/L *	1996 to 2012	Microdilution Assay (CLSI M27-S4)	>4 mg/L	[30]
Spain	Candidemia/intra-abdominal candidiasis	35	NR	Fluconazole ≥4 mg/L *	2011 to 2013	Microdilution Assay (CLSI)	>4 mg/L	[31]
Spain	Candidemia	33	NR	Fluconazole ≥ 32 mg/L	January 2006 to December 2015	Microdilution Assay (CLSI M27-S3)	>32 mg/L	[32]
Spain	Invasive candidiasis	90	4.1 1.1 1.1	Fluconazole [NR] Micafungin 2 mg/L Anidulafungin 1 mg/L	NR	EUCAST 7.3.1 microdilution	NR >2 mg/L >1 mg/L	[33]
Spain	Candidemia in Solid Organ Transplant Recipients	13	NR	Fluconazole ≥ 4 mg/L *	CANDIPOP Study—May 2010 to April 2011 CANDI—Bundle Study—September 2016 to February 2018			[34]
Spain	Candidemia	86	4.0 4.0 4.0 4.0	Isavuconazole 1–4 mg/L Fluconazole ≥ 64–128 mg/L Voriconazole 0.5–8 mg/L Posaconazole 1–2 mg/L	January 2007 to September 2017	EUCAST E. def 7.3.1	>1 mg/L >64 mg/L >0.5 mg/L >1 mg/L	[35]
Mexico	Candidemia	30	6.7 3.3 3.3	Fluconazole [NR] Micafungin [NR] Caspofungin [NR]	June 2008 to July 2014	CLSI—M27-S4	NR NR NR	[36]
Mexico	Oral	16	18.7 12.5 31.2 18.7	Miconazole [NR] Ketoconazole [NR] Itraconazole [NR] Fluconazole [NR]	NR	Diffusion CLSI M44-A	NR NR NR NR	[37]
Mexico	Oral in children with HIV	5	80.0	Fluconazole 8 mg/L *	2014	CLSI Microdilution Assay	>4 mg/L	[38]
Mexico	Esophageal candidiasis	2	100 50.0 50.0	Itraconazole ≥ 4 mg/L Fluconazole 32 mg/L Posaconazole 8 mg/L	NR	CLSI M27-A3 Microdilution Assay	NR NR NR	[39]
Paraguay	Candidemia	25	8.0	Fluconazol ≥ 64 mg/L	2010–2018	CLSI M60	>64 mg/L	[40]
Peru	Candidemia/Invasive candidiasis	8	25.0 37.5	Fluconazole [NR] Voriconazole [NR]	February 2018 to May 2019	CLSI M44-A2	NR NR	[19]
Dominican Republic	Candidemia	6	NR	Fluconazole [NR]	January 2017 to December 2018	NR	NR	[41]

MIC: minimum inhibitory concentration; NR: not reported; * Sensititre yeast considered resistance ≥ 4 mg/L.

## Data Availability

Not applicable.

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
