# Peer review of "Antifungal Resistance in Clinical Isolates of Candida glabrata in Ibero-America"

_jof, 2021, doi:10.3390/jof8010014_

Round 1

Reviewer 1 Report

I read with interest this Review entitled “Antifungal resistance in clinical isolates of Candida glabrata in Ibero-America”. The authors pointed out the scarce information available in Ibero-America when compared to other geographical regions. Therefore the topic is of interest.

Would like the authors to improve their manuscript by adding in Table 1: the year of the study (not the publication year); the AFST method; and the resistance breakpoint used for each antifungal and study.

They should discuss the influence of the breakpoint definition on the reported resistance rate. If possible the authors should describe and analyse the reported MIC50 and MIC90 in each study.

Author Response

Comments and Suggestions for Authors

First of all, I would like to thank the reviewer 1 for the work they have done in correcting our manuscript, which has been very helpful. Below we have answered the questions and addressed the corrections suggested to us:

Reviewer: 1

Comments to the Author

I read with interest this Review entitled “Antifungal resistance in clinical isolates of Candida glabrata in Ibero-America”. The authors pointed out the scarce information available in Ibero-America when compared to other geographical regions. Therefore the topic is of interest.

Would like the authors to improve their manuscript by adding in Table 1: the year of the study (not the publication year); the AFST method; and the resistance breakpoint used for each antifungal and study.

Authors' reply

Three columns were added to table 1, one with years studied, ASFT method and resistance breakpoint

Comments to the Author

They should discuss the influence of the breakpoint definition on the reported resistance rate. If possible, the authors should describe and analyse the reported MIC50 and MIC90 in each study.

Authors' reply

The influence of the definition of the cut-off points on the reported resistance rate was discussed in lines 247-257. Since only five of the articles which were included in this study report MIC50 and MIC90, the suggested analysis was not included.

Reviewer 2 Report

The article by Hererra et al, describes incidence of antifungal resistance in C. glabrata clinical strains isolated from patients from ibero-american region. The review article summarized the data published in literature between 2015-2020 regarding incidence of antifungal resistance in C. glabrata in ibero-american region. Here are my comments:

1) The review article should state the known C. glabrata drug resistance mechanisms summarizing the findings in the following:
https://pubmed.ncbi.nlm.nih.gov/32441527/

https://pubmed.ncbi.nlm.nih.gov/32526921/

2)Are mechanisms of antifungal resistance region specific as well? Please clarify

3)Are there any reports as to the drug resistance mechanisms in these strains? If so that need to be analyzed to make the paper more relevant.

4)A pie chart showing which drug is most prone to resistance in the ibero-american isolates will greatly improve the review.

5)All genus names in the introduction needs to be italicized (Lines 42,46 and 82)

6)Is cross resistance to multiple drugs common in Ibero-american region?

Author Response

Review 2.

First of all, I would like to thank the reviewer 2 for the work they have done in correcting our manuscript, which has been very helpful. Below we have answered the questions and addressed the corrections suggested to us:

Comments to the Author

The article by Hererra et al, describes incidence of antifungal resistance in C. glabrata clinical strains isolated from patients from ibero-american region. The review article summarized the data published in literature between 2015-2020 regarding incidence of antifungal resistance in C. glabrata in ibero-american region. Here are my comments:

The review article should state the known C. glabrata drug resistance mechanisms summarizing the findings in the following:
https://pubmed.ncbi.nlm.nih.gov/32441527/
https://pubmed.ncbi.nlm.nih.gov/32526921/

Authors' reply

Resistance mechanisms were included based on the articles suggested by the reviewer No. 1 (Line 85-108)

Comments to the Author

2) Are mechanisms of antifungal resistance region specific as well? Please clarify

Authors' reply

No, in fact, antifungal resistance mechanisms have been studied in other regions of the world.

Comments to the Author

3) Are there any reports as to the drug resistance mechanisms in these strains? If so that need to be analyzed to make the paper more relevant.

Authors' reply

No, none of the reviewed articles specifies the analysis of the resistance mechanisms involved.

Comments to the Author

4) A pie chart showing which drug is most prone to resistance in the ibero-american isolates will greatly improve the review.

Authors' reply

The graph was made

Round 2

Reviewer 1 Report

The authors adequately adressed my remarks.

Author Response

There are not comments and suggestions for authors Thank you